# Real-time tracking of the intramolecular vibrational dynamics of liquid water
Gaia Giovannetti[1], Sergey Ryabchuk[1,2], Ammar Bin Wahid[1], Hui-Yuan Chen [3], Giovanni Batignani [4], Erik P. Månsson [1], Oliviero Cannelli [1], Emanuele Mai [4], Andrea Trabattoni[1,5], Ofer Neufeld[6], Angel Rubio [1,7], Vincent Wanie [1], Hugo Marroux[8] ✉, Tullio Scopigno [4] ✉, Majed Chergui[3,9] ✉ & Francesca Calegari [1,2,10] ✉

Water's polarity and hydrogen-bond network give rise to its unique chemical and biochemical behaviour. Its vibrational motions, occurring on a few-femtosecond timescale, govern ultrafast energy transfer within the hydrogen-bond network. However, direct real-time observation of these motions has remained elusive due to the extreme temporal resolution required. Here, we investigate the ground-state vibrational dynamics of liquid water initiated by a sub-5 fs near-infrared (NIR) pump pulse via Impulsive Stimulated Raman Scattering (ISRS). Using few-fs ultraviolet (UV) probe pulses transmitted through a 5 µm-thick liquid jet, we monitor the coherent vibrational wave packet dominated by the OH stretch mode, exhibiting a 10 fs oscillation period and a 25 fs damping time. These results reveal the rapid dephasing of the OH stretch mode preceding its relaxation through coupling to the bending vibrations, highlighting the importance of intermolecular couplings of liquid water in the high frequency vibrational dynamics.

The hydrogen bond network of liquid water plays a crucial role in determining its unique properties as a solvent in chemistry and biology. A striking example is how its dynamic behaviour is significantly affected by salts belonging to the Hofmeister series[1]. Indeed, these salts modify the solvation properties of liquid water surrounding proteins, as the hydrogen bonds mediate the interactions between the salts and the macromolecules[2,3]. Additionally, compared to the gas phase, the strong intermolecular interactions severely affect reaction rates taking place in aqueous solutions. In particular, the excess of vibrational energy in a chemical reaction can rapidly be dissipated in the liquid phase, making such reactions effectively irreversible[4-6]. In addition, intramolecular electronic energy relaxation was noted to take place at extremely short, sub-vibrational timescales via high frequency molecular modes[7]. While most of this energy is rapidly funnelled into lower-frequency modes within the solute, depending on the coupling with the solvent, part of it may well be transferred to the latter on ultrashort time scales. This energy transfer is particularly evident in the ground electronic state where, due to the lower density of states, the energy relaxation involves coupling to solvent modes[8].

The vibrational infrared and Raman spectrum of bulk liquid water in the ground electronic state is dominated by the bending modes and the OH stretching, between 1000 and 4000 cm$^{-1}$. There is a substantial broadening of the OH stretch vibrational bands by several hundreds of wavenumbers as a direct consequence of the extremely high sensitivity to the molecular environment. More specifically, the line broadening typical of the liquid phase reflects a stronger intermolecular coupling, resulting in a faster decay and a larger spatial delocalization of the vibrational excitation. Indeed, while in the gas phase the energy transfer relies mostly on intramolecular mode coupling, both intra- and intermolecular couplings play an essential role in the condensed phase. In the case of bulk liquid water, the OH stretch modes are believed to play an important role in the vibrational energy relaxation dynamics. Both experimental and theoretical studies[9-15] have demonstrated the existence of different parallel pathways of vibrational energy transfer involving the OH stretch modes. At the intramolecular level, the stretch modes are coupled to the bend mode and their overtones via anharmonic interactions[16]. At the intermolecular level, the coupling between the OH stretching modes of different water molecules is known to be responsible for

[1]Center for Free-Electron Laser Science CFEL, Deutsches Elektronen-Synchrotron DESY, Notkestr. 85, Hamburg, Germany. [2]The Hamburg Centre for Ultrafast Imaging, Universität Hamburg, Luruper Chaussee 149, Hamburg, Germany. [3]Lausanne Centre for Ultrafast Science (LACUS), Ecole Polytechnique Fédérale de Lausanne, ISIC, FSB, Station 6, Lausanne, Switzerland. [4]Dipartimento di Fisica, Università di Roma "La Sapienza", Roma, Italy. [5]Institute of Quantum Optics, Leibniz Universität Hannover, Hannover, Germany. [6]Schulich Faculty of Chemistry, Technion—Israel Institute of Technology, Haifa, Israel. [7]Max Planck Institute for the Structure and Dynamics of Matter, Luruper Chaussee 149, Hamburg, Germany. [8]Laboratoire Interactions, Dynamiques et Lasers, CEA-Saclay, Gif-sur-Yvette, France. [9]Elettra - Sincrotrone Trieste S.C.p.A., S.S. 14 Km 163, 5 in Area Science Park, Trieste, Italy. [10]Physics Department, Universität Hamburg, Luruper Chaussee 149, Hamburg, Germany. ✉e-mail: hugo.marroux@cea.fr; tullio.scopigno@uniroma1.it; majed.chergui@elettra.eu; francesca.calegari@desy.de

a quasi-instantaneous delocalization of the vibrational excitation across several molecules[14,15] and for the dissipation of vibrational energy to happen on an ultrafast time scale, namely below 100 fs[14,15,17–19]. Frequency-resolved measurements report for the OH stretching mode a band centred around 3500 cm$^{-1}$ (10 fs periodicity) with a full width at half maximum of 435 cm$^{-1}$, which approximately corresponds to a decay time of 24.4 fs[9]. The latter reflects a combination of population decay, quantum decoherence and frequency dephasing, corresponding to fluctuations of the energy gap. Additionally, photon echo studies of the OH stretch decoherence for HOD in $D_2O$ have reported time scales between 90 and 130 fs for the homogeneous dephasing time[14]. These echo decay times include contributions from both quantum decoherence and dephasing due to fluctuations in the OH vibrational transition energy. In the case of liquid water, these decays are expected to become significantly faster, therefore requiring few-femtosecond time resolution to be monitored in real time. As a matter of fact, pump-probe measurements on liquid water have so far been limited to a time resolution of a few tens of femtoseconds[14,15,17,18], and a real time investigation of vibrational dynamics is still missing, which could shed light into yet faster dynamical phenomena such as energy transfer between vibronic and electronic degrees of freedom.

Here, we present a time-resolved study of the OH stretching motions initiated in the ground state (GS) of liquid water by a sub-5 fs NIR pump pulse. The rapid evolution of a vibrational wave packet, created by impulsive stimulated Raman excitation in a thin (~5 μm) liquid jet of water, is then probed with few-femtosecond time resolution by monitoring the transmission of an ultrabroadband (220–320 nm) UV probe pulse[20] (see Fig. 1a and "Methods"). The experimental data show the evolution of a wave packet of the OH stretch mode that oscillates at a period of ~10 fs and it decays within ~25 fs. The latter is agreement with the data retrieved from Raman spectroscopy. The ability to visualize the wave packet of solvent modes in real-time suggests a strategy to investigate impulsive solute-to-solvent energy transfer in a wide range of solvents, or alternatively, the coupling of ground state vibrational modes of the solute, induced by ISRS, to the solvent.

## Results

Figure 1a shows a sketch of the experimental setup: a sub-5fs NIR pulse is focused on a thin liquid jet of water and excites the sample via ISRS[21,22]. A time-delayed ultrabroadband UV pulse is then used to probe the relaxation dynamics taking place within few femtoseconds[23] (see "Methods" and

Supplementary Note 2). Figure 1b shows the differential probe spectrum recorded over 150 fs. In the temporal region in which the pump and the probe do not overlap, the transient absorption signal exhibits fast modulations, which decay rapidly. Additionally, within the bandwidth of the UV probe pulse, we can discern two regions where the signal oscillates with opposite phase between positive and negative values.

In Fig. 2, we plot the transient signals extracted by integrating the portions of the spectrum centred at $\lambda_{blue} = 238$ nm (Fig. 2a) and $\lambda_{red} = 290$ nm (Fig. 2b) over a 30 nm region, and after background subtraction. This background, which is slowly varying in wavelength and time, results from the combination of spurious non-linearities and stray light. An out-of-phase oscillatory dynamics is clearly visible. To improve the contrast, we scanned a shorter temporal range (50 fs) with the finer step of 1.5 fs (see Fig. 2c). The π phase difference between the oscillations detected at the two opposite sides of the probe spectrum can be rationalized considering the energy level diagrams describing the impulsive Raman process (Fig. 2d, e). Two pathways, labelled as A (red side of the spectrum) and B (blue side of the spectrum), concur to the ISRS signal generation and can respectively be assigned to the $|g\rangle\langle g|$ and the $|g\rangle\langle g'|$ coherences, stimulated by the pump pulse and subsequently scattered off by two different colours of the probe (i.e., red or blue shifted by the vibrational energy, depending on the diagram). As a result, the Raman signal is emitted at a probe wavelength shifted by a vibrational quantum with respect to the probing wavelength. Since the quantum path associated with A has opposite phase compared to that associated with B, the two oscillatory patterns exhibit a relative π phase shift, which inevitably leads to a quenching of the signal at the central frequency of the probe spectrum (see Discussion for further details).

A damped harmonic oscillator (DHO) analysis of the signals reported in Fig. 2a and b reveals dynamics characterized by a single decay of 25.0 fs (see Supplementary Note 1 for the model), which matches with the lifetime of 24.4 fs associated to the bandwidth of the ground state OH stretching band in frequency-domain experiments[9] (Fig. 3a). From the Fourier analysis of both the blue (Fig. 3b) and red (Fig. 3c) portions of the spectrum, we identify a main peak at 3335 cm$^{-1}$, corresponding to the OH stretching mode, confirming that this vibrational mode dominates the coherent wave packet dynamics. A weaker contribution around 2100 cm$^{-1}$ can also be observed, which corresponds to the combination of the bending and the librational modes[24]. The Fourier analysis of the shorter scans illustrated in Fig. 2c also reveals a main feature between

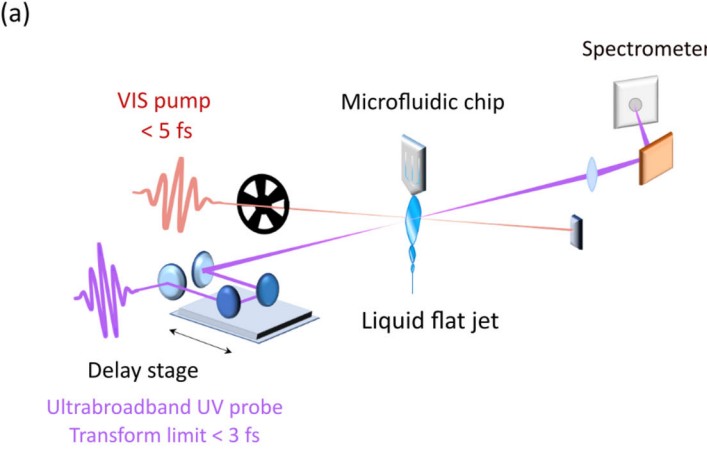

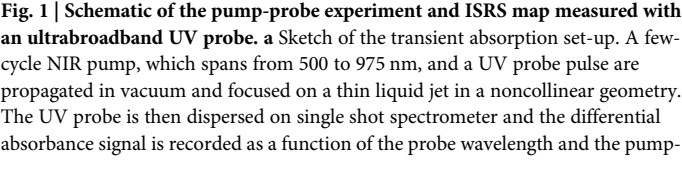

**Fig. 1 | Schematic of the pump-probe experiment and ISRS map measured with an ultrabroadband UV probe. a** Sketch of the transient absorption set-up. A few-cycle NIR pump, which spans from 500 to 975 nm, and a UV probe pulse are propagated in vacuum and focused on a thin liquid jet in a noncollinear geometry. The UV probe is then dispersed on single shot spectrometer and the differential absorbance signal is recorded as a function of the probe wavelength and the pump-

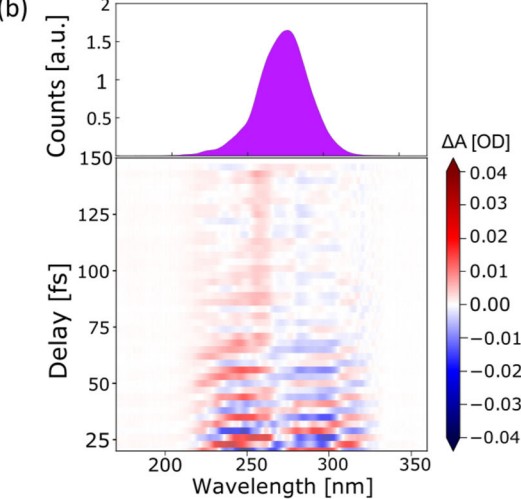

probe time delay. **b** UV probe static spectrum (top) and colour map of the measured transient absorption signal as a function of the pump-probe time delay (bottom). The large bandwidth of our UV pulses allows to simultaneously probe the dynamics in different spectral regions. In particular, the blue (left) and red (right) sides of the transient absorption exhibit out-of-phase oscillations between positive and negative values.

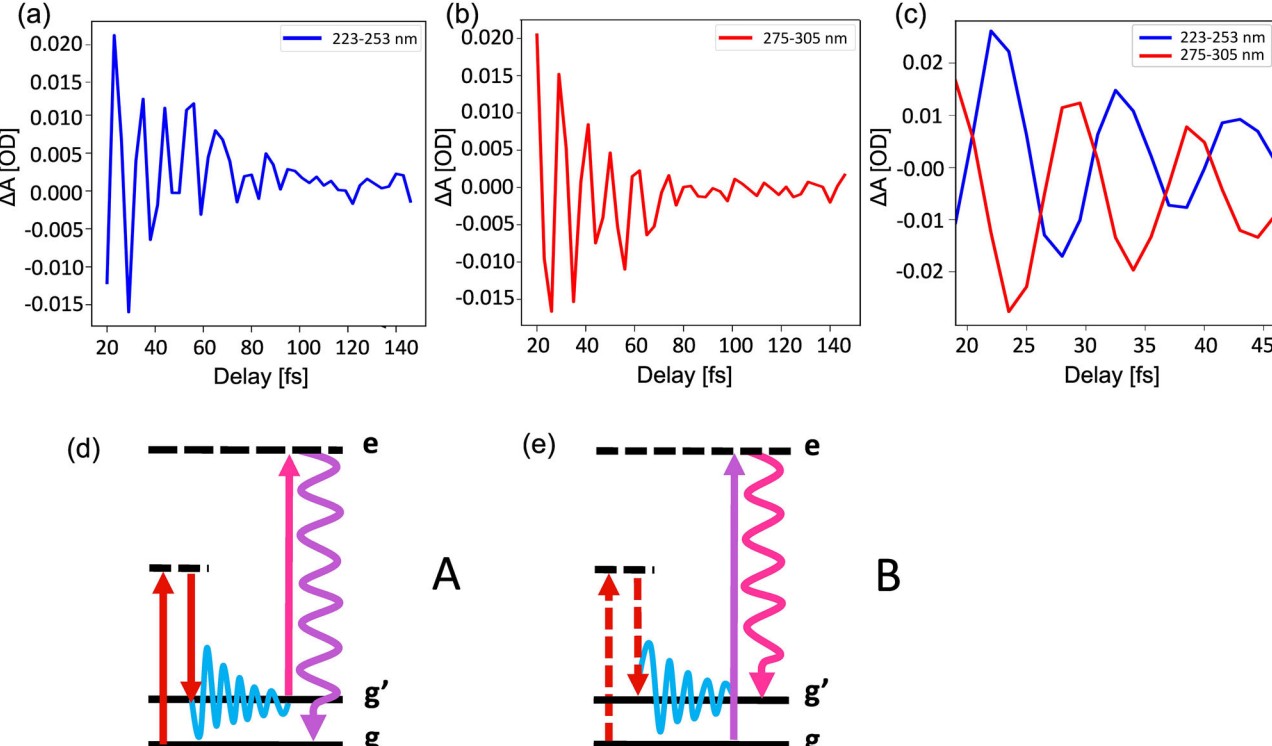

**Fig. 2 | Transient signals of liquid water. a, b** Time traces extracted from the blue and red side of the spectrum in Fig.1b. The signal is averaged over the 223–253 nm and 275–305 nm wavelength range, respectively. **c** Comparison between the blue and red traces, from additional measurements recorded with a 1.5 fs time delay step. The use of a finer time step allows to visualize the $\pi$-phase difference between the two sides of the spectrum. **d, e** Energy level diagrams describing the two spectroscopic contributions A and B, respectively, which form the ISRS signal consisting in two traces centred at different wavelengths and separated by a central node, as described in the main text. $g$ and $g'$ represent the $v = 0$ and $v = 1$ vibrational levels of the OH stretch oscillator in the electronic ground state of the system, while $e$ represents an electronic virtual state.

3000 and 4000 $cm^{-1}$ (see Supplementary Note 4 and Supplementary Fig. 7). All these vibrational dynamics result from the non-resonant ISRS process taking place in the GS of liquid water. We remark that the spontaneous Raman and ISRS cross sections depend differently on the thermal population of a given mode, due to the coherent nature of the latter[25]. Moreover, the ISRS intensity ratios are modulated by the temporal and spectral properties of the beams, and may differ from the spontaneous Raman spectrum[26].

It is worth to stress that the probing scheme employed here is non-resonant because the ground state electronic absorption of liquid water into the first (purely repulsive) electronic $A^1B_1$ excited state starts above 7 eV[27]. Even when populating up to 4-5 vibrational quanta (amounting to an energy $\leq 2$ eV), which is unlikely by ISRS with the present pump-pulse spectral content, is it not possible to reach the $^1B_1$ state with our probe pulse. Importantly, the ability to investigate high-frequency modes via nonresonant ISRS strictly depends on the temporal and spectral characteristics of the pump and probe pulses. The non-resonant nature of the NIR pump (Fig. 2d) guarantees that the vibrational coherence is solely generated in the electronic GS of the system as a result of a Raman excitation, as demonstrated by the power scaling measurements (Supplementary Note 7). Additionally, the duration of the pump pulse must be shorter than the period of the vibrational mode, in order for the coherence to be efficiently stimulated. For this reason, the sub-5 fs duration of the pump pulses is a key element in this experiment, since it allows to stimulate a coherence involving a high-energy mode like the OH stretching and supports a bandwidth which allows the vibrational coherence to extend over two, possibly, three vibrational levels of the OH stretch mode (Fig. 2d). The broad band nature of the UV probe pulse allows for high-temporal resolution while ensuring the detection of high vibrational frequencies with a relatively high signal-to-noise ratio. As we will show below, the presence of two distinct regions and

the energy-dependent phase of the oscillations are strongly affected by the chirp of the UV probe pulses.

## Role of the UV chirp

To better investigate the vibrational dynamics in liquid water, the spectral shape and temporal chirp of the UV probe pulses were tuned. In particular, the transient spectral signal was recorded for three different values of the UV probe chirp, which correspond to the three different UV spectra shown in Supplementary Fig. 3. Since a measurement of the UV spectral phase is challenging for ultrabroadband UV pulses, the chirp was changed over the three different values C1, C2, and C3. The values of the UV pulse chirp retrieved from the experimental data are $-3.8$ $fs^2$, $4.2$ $fs^2$ and $2.8$ $fs^2$ for C1, C2, and C3, respectively (see Supplementary Note 3 for details).

Figure 4a–c shows three time-dependent transient absorption signals acquired for the three different chirp values C1, C2, and C3, after background subtraction and removal of the coherent artefact (pink area)[26,28]. For the UV chirp value C1 (see Supplementary Fig. 4 for more details), the time-dependent signal shows a negative slope, according to the relative negative sign of the UV chirp. By tuning the chirp over the C2 and C3 values, one can observe a change in the sign of the slope, corresponding to a change of the sign of the UV chirp. Interestingly, in the measurements corresponding to chirp values C1 and C2 the central node observed in Fig. 1b is absent, while a clear quenched signal can be observed in correspondence of the chirp value C3 (same UV pulse used also for the measurement reported in Fig. 1b). These results emphasize the role of the probe chirp when determining the features of the ISRS signal in the probe spectrum. Figure 4g–i shows the transient signal extracted for different wavelengths in the probe spectrum for each of the three measurements. While for C1 (C2) the phase of the ISRS experimental signal monotonically decreases (increases) when going from the blue side to the red side of the spectrum, for C3 a sudden phase jump is

**Fig. 3 | Free-induction decay of the OH stretch mode of water and its Fourier transform.** **a** Lorentzian fit (pink solid line) of the experimental transient signal extracted from the blue side of the spectrum (red dotted line). The coherence decays according to a single exponential (orange dashed line) dynamic characterized by a time constant of 25.0 ± 4.4 fs. The blue shaded area represents the standard deviation over two delay scans. **b, c** Fourier spectra of the blue and red side traces, after integration on a 30 nm bandwidth (223–253 and 275–305 nm, respectively). The main feature consists in a large band extending between 3000 and 4000 cm⁻¹, corresponding to the OH stretch mode of liquid water.

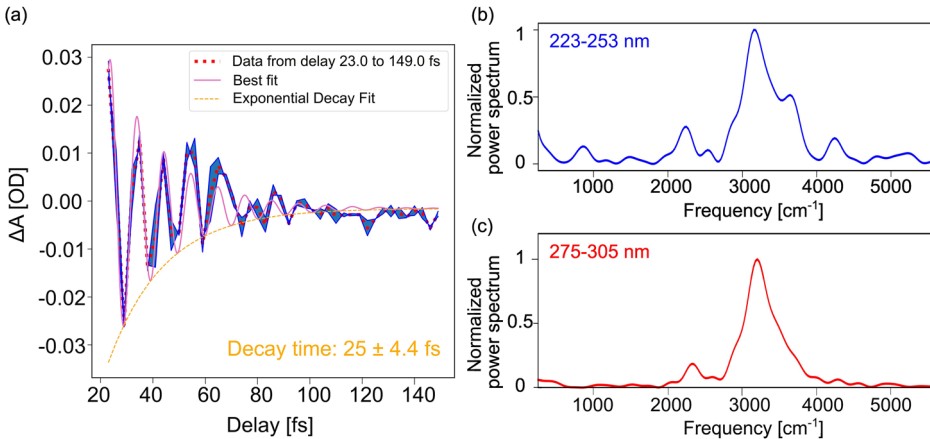

observed, in agreement with a quenched signal amplitude at the central wavelength of the probe spectrum.

## Discussion

As mentioned above, following the diagrammatic description of the density matrix formalism[29], the blue and red side traces can be interpreted as the result of two distinguished spectroscopic contributions, described by energy level diagrams A and B (Fig. 2d, e), respectively. From a physical perspective, the coherent vibrational excitations triggered by the pump beam are scattered off by competing non-resonant Stokes and anti-Stokes transitions, with opposite phases[23,30]. The ISRS signal arises hence from the interference of these two quantum pathways, responsible for time traces oscillating with opposite phases, which critically cancel out in the central region of the spectrum. In both diagrams, the non-resonant NIR pump prepares a coherence in the ground state of the system via an intrapulse Raman excitation. After a tuneable pump-probe delay time, the rapid evolution of the vibrational wave packet is then probed by the different components of the ultrabroadband UV pulse, which scatters off the vibrational coherence leaving the system in a population state. The relative phase shift of the two contributions can be traced back to the different pump-matter interaction taking place in the blue and the red side. Indeed, while on the blue side the preparation of the coherence involves the interaction of the NIR pump with the ket of the molecular density matrix (Fig. 2d, red solid arrows), on the red side the pump interacts with the bra of the system (Fig. 2e, red dashed arrows), leading to coherence with frequency $\omega_{g'g}$. The two vibrational coherences created in pathways A and B, $|g'\rangle\langle g|$ and $|g\rangle\langle g'|$, evolve in time as $e^{i\omega_{g'g}t}$ and $e^{-i\omega_{g'g}t}$, respectively, resulting in a $\pi$-phase shift between the signal oscillating on the blue and red sides of the spectrum (see Supplementary Note 5 for more information). As a result, the signal is strongly quenched at the central probe wavelength due to the destructive interference between the A/B processes.

Interestingly, the ISRS signal is generated by an interaction with probe spectral components that are red and blue shifted with respect to the probe detection wavelength; hence, introducing a chirp (a delay between the probe different colours)[31] offers the chance to introduce a relative phase between the A/B contributions. In this experiment, this is exploited to enhance the ISRS signal and prevents its total cancellation around the central wavelength of the probe. Additionally, the possibility of fine tuning the chirp of the probe pulses results in the ability to modulate the Raman signal and therefore to modulate the cross section of the experimental signal by selectively enhancing a desired vibrational mode[32].

Figure. 4d–f shows the simulated ISRS signal for the three UV chirp values. The ISRS signal in each dataset can be modelled according to the fit equations A and B (see "Methods"), which describe the oscillating signal in the blue and red sides of the spectrum, respectively. The frequencies, intensities, dephasing values of the modes are extracted by

fitting the spontaneous Raman spectrum of liquid water[33] with the minimum number of components that correctly reproduce both spontaneous Raman and ISRS data (see Supplementary Note 6); the inhomogeneous spectral broadening contributions are modelled as Gaussian decays. Each ISRS dataset is fitted with 6 free parameters: the pump amplitude, the time zero and four terms accounting for the linear and nonlinear chirp of the probe. Initial values of the chirp parameters can be estimated performing a polynomial fit of the coherent artefact and used to simulate the ISRS oscillating signal. Importantly, the fit of the oscillatory signal can be conducted according to different approaches (see Supplementary Note 6). The first approach consists in fitting the ISRS signal after fixing the value of the chirp parameters. In this case, the agreement between the experimental and modelled signal, averaged on the three maps and evaluated as the ratio between the residuals and the square modulus of the data amplitude, is 26%. In the second approach, the value of the chirp parameters extracted from the calibration of the coherent artefact are instead free to finely recalibrate during the fit of the ISRS oscillations, to achieve a better agreement between the experimental signal and the model (Fig. 4). The recalibration of the chirp improves the numerical average agreement between the experimental and modelled signal with respect to a fit performed with fixed chirp parameters by 12%, stressing the importance of knowing the dependence of the signal on the chirp of the UV probe. The values of the linear chirp of the probe extracted for the three data sets are −3.8, 4.2, and 2.8 fs² for C1, C2 and C3, respectively (see Supplementary Figs. 4, 5 and 6). All these values are close to the chirp values necessary to enhance the high-frequency region of the Raman spectrum. This fact, along with a significant difference in cross section between the high and low-frequency modes of liquid water, explains why the ISRS signal is dominated by the OH stretching vibration, while the other Raman active modes appear quenched in the recorded maps. Figure 4g–i compares the different experimental transient signals at selected probe wavelengths with the corresponding simulated ISRS signals for the three different data sets. The coherent artefact region has been truncated, and the data have been vertically offset for improving visibility (complete datasets are reported in the Supplementary Note 8). The experimental data and the modelled signals show excellent agreement for each probe chirp value and across all selected wavelengths.

Our results provide a glimpse into the ultrafast vibrational dynamics of liquid water, highlighting the role of the liquid environment in determining the time scale on which the dissipation of excess vibrational energy and elastic dephasing occurs, complementing experimental approaches with lower temporal resolution. Previous studies had focused on the relaxation pathway of the water vibration, and pointed to the fact that the Fermi resonance between the stretch modes and the first overtone of the bend mode plays a central role in the relaxation of the OH stretch mode[34,35]. Furthermore, ultrafast broadband two-dimensional (2D) infrared

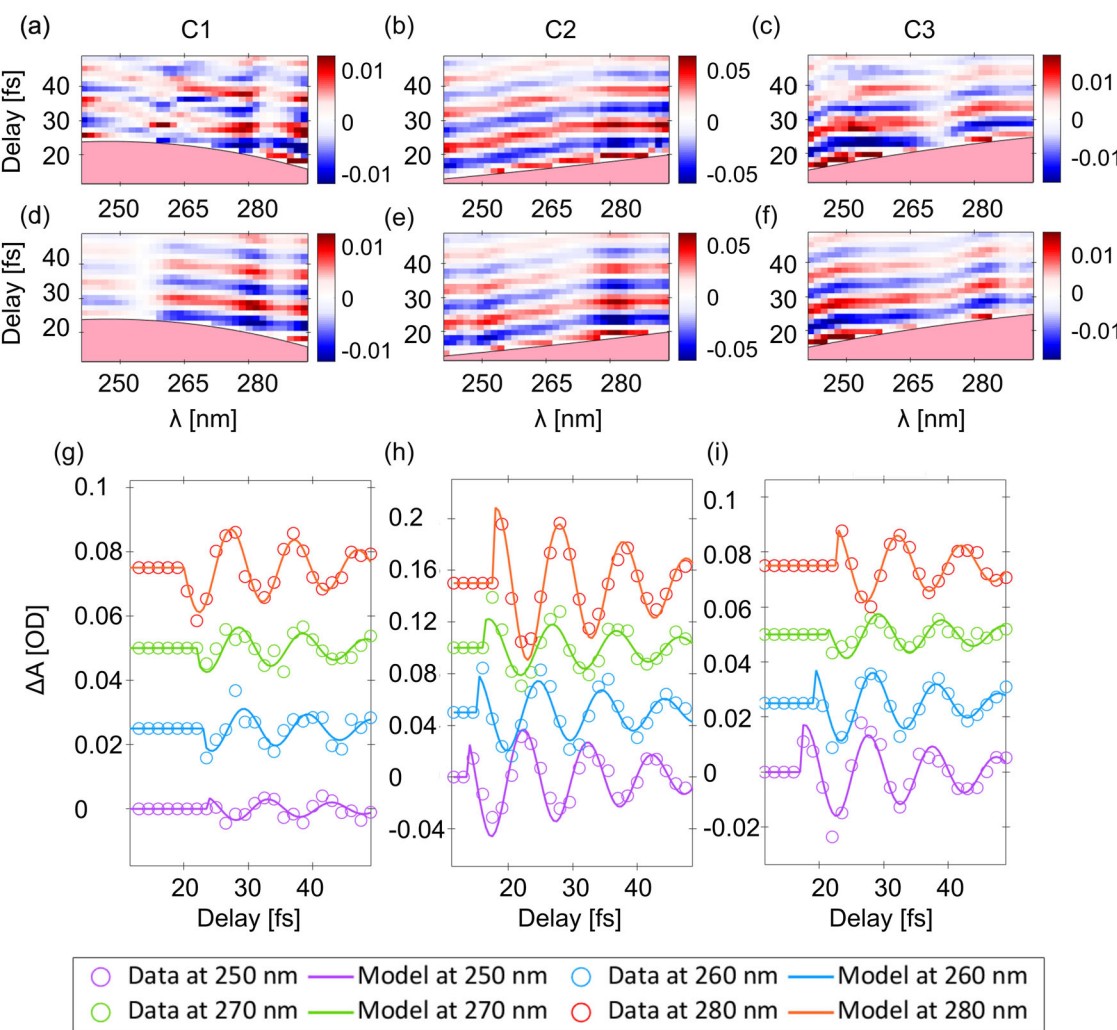

**Fig. 4 | Comparison between the ISRS experiment and model. a–c** Colour maps of the experimental signal for three different values of the UV chirp, C1 (−3.8 fs²), C2 (4.2 fs²), and C3 (2.8 fs²). The three scans were recorded with the same NIR pump pulse and by preserving the same UV probe pulse energy. **d–f** Modelled ISRS maps.

The values of the chirp are initially extracted by fitting the coherent artefact and subsequently recalibrated in order to optimize the agreement between the model and the experiment (see Supplementary Information). **g–i** transient absorption extracted for different probe wavelengths and for the three different values of the UV chirp.

spectroscopy[15,36] concluded that ultrafast memory loss occurs under 50 fs, presumably due to intermolecular motions preceding vibrational relaxation, mediated by the Fermi resonance. Indeed, Ashihara et al.[35] reported that upon OH stretch excitation, the OH bend oscillator shows an instantaneous response due to a Fermi resonance between the v = 1 stretch and v = 2 bend levels. The coupled states decay with a 200-fs lifetime via the OH bending and librational modes. Somewhat similar trends were reported by Lindner et al.[34]. Here, we reveal that at ultrashort time scales, the OH stretch vibration undergoes a dephasing in 25 fs, which however, is unlikely to be due to the Fermi resonance coupling as its value (~30 cm⁻¹)[9] corresponds to much slower time scales. We therefore conclude that the dephasing time we report is predominantly due to elastic dephasing via intermolecular couplings, in line with the <50-fs memory loss invoked by Cowan et al. The question remains as to how these intermolecular couplings proceed. In a recent study, using ultrafast electron diffraction, Yang et al.[37] investigated the structural dynamics of liquid water at ultrashort times, upon direct IR excitation of the OH stretch band and probing with ultrashort MeV electron pulses. They concluded that the first intermolecular structural response is an OO contraction by ~0.04 Å in the first coordination shell that occurs in ~80 fs, based on the consideration that the OH stretching motion strengthens the H-bonding, while the bending and librational modes weaken it[38]. The OO contraction they observe reflects an intermolecular structural reorganization (the motion of the oxygen atoms) prior to the relaxation into the

bending overtone mediated by the Fermi resonance coupling, which involves only the motion of light hydrogen atoms. Our observations provide a lower limit and confirm that an ultrafast event is indeed operative and is driven by inter- rather than intramolecular couplings.

In conclusion, we demonstrate the unique experimental capabilities to generate, via ISRS, a ground state vibrational wave packet of liquid water and probe its evolution in real-time with few-fs temporal resolution. Both pump and probe are off-resonance, which is an important aspect of the generalization of this approach to a wide variety of solvents and ultimately also solids[39]. Furthermore, this suggests a strategy to investigate the energy relaxation of the ground state of solute molecule, e.g. by impulsively generating vibrational wave packets therein via ISRS and reading out their response via the solvent vibrational modes. The interpretation of the results is benchmarked by a perturbative expansion of the sample density matrix to model the experimental signals, which rationalizes the chirp dependence of the ISRS signal as an interference effect between concurring interaction pathways. Importantly, our results show that chirped broadband probe pulses can be employed to selectively unveil the vibrational dynamics of high-frequency modes in liquids. This latest development is particularly significant in the context of impulsive electronic excitation of the solute, which is subsequently followed by a rapid relaxation process that likely involves high-frequency intramolecular modes.

## Methods

### Experimental setup

The experiments were carried out with a multipass Ti: Sa amplifier (Femtopower, Spectra Physics), which delivers 25 fs, 10 mJ, CEP-stable laser pulses at a 1 kHz repetition rate. A portion of the beam (6 mJ) is coupled into a stretched hollow-core fibre operating with a pressure gradient up to 1.7 bar He. The output pulses are post-compressed to 4.5 fs using a chirped mirror compressor (few-cycle inc.). In the beamline[20], the beam is split into two arms, which serve to prepare the pump and probe beams, both of which propagating in high vacuum ($<10^{-4}$ mbar). The pump NIR pulse (with energy of approximately 80 $\mu$J) is centred at 1.635 eV (758 nm) and has a bandwidth of 1.208 eV (Supplementary Fig. 1). It is focused onto the experimental region by a gold-coated toroidal mirror at grazing incidence and a motorized iris is used to adjust the peak intensity at the experimental point to three values: $2.4 \times 10^{12}$, $4.1 \times 10^{12}$, and $9.1 \times 10^{12}$ W cm$^{-2}$.

The probe is generated by focusing the second replica (450 $\mu$J) into a laser machined glass cell filled with Ar to generate few-cycle UV pulses via third-harmonic generation (THG)[40]. The UV pulses are subsequently separated from the fundamental NIR radiation using two Silicon mirrors at Brewster's angle. The resulting UV pulses exhibit a broadband spectrum covering the 3.9–5.6 eV spectral range (Supplementary Fig. 2) supporting a 2.7 fs transform-limited pulse duration. The spectral dispersion of the UV pulses is tuned by inserting a variable amount of glass in the path of the generating NIR beam, each associated with a different UV spectrum (Supplementary Fig. 3), which support a transform-limited pulse duration between 1.9 fs and 2.7 fs. The effective pulse duration, which varies between 5.8 and 10 fs, has been retrieved from the experimental data for the three acquisitions in which the chirp of the probe is varied (see Supplementary Note 3, MEP_L_dmmc2). In this experiment, the UV pulse energy varies between 40 and 50 nJ. The UV pulses are focused onto the target using a 90 cm focal length spherical mirror and are recombined with the pump in a noncollinear geometry with an angle of 0.6 degrees. The amplitude of the NIR pump is modulated by a mechanical chopper operating at 500 Hz. After the interaction with the sample, the probe is dispersed on a spectrometer and the dynamics of the system can be followed by monitoring the evolution of the transient differential absorption of the probe on a single-shot basis.

The sample is delivered through a thin flat jet generated by a microfluidic chip device[41]. The chip operates in the colliding configuration, producing a stable jet with 5 $\mu$m thickness at the centre of the leaf. The jet operates in vacuum and at room temperature, while the pressure in the chamber is maintained slightly above the vapour pressure of water (23 mbar at 20 degrees) to prevent the jet from freezing due to evaporative cooling.

### Theory

The ISRS signal has been interpreted using a diagrammatic semiclassical approach, based on a perturbative expansion of the density matrix, representing the matter. According to this model, the nonlinear polarization responsible for the ISRS process can be evaluated through a third-order expansion of the molecular density matrix in terms of the electric fields (the Raman pump and the probe). The various pathways contributing to the total spectroscopic response can then be depicted using Feynman diagrams[42] (Fig. 2d, e). This approach allows for the dissection of different spectroscopic pathways contributing to the nonlinear response as a function of the probe wavelength. The coherent part of the signal is modelled using Eqs. (1) and (2), which describe the oscillatory signal in the blue and red side of the spectrum, respectively. A detailed description of the theoretical framework is presented in Supplementary Note 5.

$$S_A(\omega, t_0) = -\Re\left[\frac{E_P^*(\omega)}{I_P(\omega)} K \frac{\widehat{I_R^{(0)}}(\tilde{\omega}_{g'g}) E_P(\omega - \tilde{\omega}_{g'g}) e^{-i\tilde{\omega}_{g'g} t_0}}{\tilde{\omega}_{eg} - \omega}\right] \tag{1}$$

$$S_B(\omega, t_0) = +\Re\left[\frac{E_P^*(\omega)}{I_P(\omega)} K \frac{\widehat{I_R^{(0)*}}(\tilde{\omega}_{gg'}) E_P(\omega + \tilde{\omega}_{gg'}) e^{+i\tilde{\omega}_{gg'} t_0}}{\tilde{\omega}_{eg'} - \omega}\right] \tag{2}$$

and

$$K = \frac{\partial \alpha}{\partial Q_{g'}} \frac{\mu_{ge}\mu_{eg'}}{\hbar^2}.$$

Equations (1) and (2) describe two classes of signals at a given probe frequency $\omega$, which are centred at $\omega_{eg}$ and $\omega_{eg}'$, respectively, and are proportional to the amplitude of the probe $E_P$ and of its complex conjugate $E_P^*$ at frequencies that are red ($S_A$) or blue-shifted ($S_B$) by one vibrational quantum $\omega_{gg'}$ relative to the measured probe frequency. Additionally, $S_A$ and $S_B$ depend on the probe intensity $I_P(\omega)$ and the Fourier transform of the pump intensity profile $\widehat{I_R^{(0)}}(\omega)$ and on the molecular characteristics via the derivative of the molecular polarizability $\alpha$ with respect to the coordinate $Q$ of the specific vibrational mode and the transition dipole moments $\mu_{ge}$ $\mu(eg')$. For an electronically off-resonant probe with transform-limited duration ($C_j=0$ in Eq. 3, Supplementary Note 5) and a flat probe spectral profile, these two signals generate time oscillations with the same amplitude but opposite phase; under these conditions, $S_A$ and $S_B$ interfere destructively, suppressing the ISRS signal[21,30].

## Data availability

The data are available from the corresponding authors upon reasonable request.

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

## Acknowledgements

H.-Y.C. and M.C. acknowledge support by the ERC Advanced Grants DYNAMOX (n° 695197) and CHIRAX (n° 101095012). F.C. acknowledges funding from Cluster of Excellence 'CUI: Advanced Imaging of Matter' of the Deutsche Forschungsgemeinschaft (DFG)—EXC 2056—project ID 390715994, the DFG—SFB-925—Project ID 170620586, the Helmholtz-Lund International Graduate School (HELIOS), the Centre for Molecular Water Science (CMWS). F.C., T.S., G.B. and E.M. acknowledge funding from the European Union under HE-GA 101131771 Lasers4EU (project ID 36977), V.W. acknowledges funding from the German Research Foundation (DFG) - project ID 545611997. O.C. acknowledges the SNSF Postdoc mobility programme under the grant agreement P500PN_214151. G.B., E.M. and T. S. acknowledge the support by EU HORIZON programme GRAPHERGIA through grant GA-101120832 and by the ERC Starting Grant 2D-PULSES (n° 101163180). H.M. acknowledges support by the ERC starting grant SATTOC (n° 101078595). A.T. acknowledges support from the Helmholtz Association under the Helmholtz Young Investigator Group VH-NG-1603, and financial support from the European Research Council under the ERC Soft-Meter no. 101076500.

## Author contributions

H.M., M.C. and F.C. conceived the experiment. G.G., S.R., A.B.W., H-Y.C., E.P.M., V.W. and H.M. carried out the experiments. G.G. and G.B. analysed the data; G.B. and T.S. performed the simulations. O.N., A.R., O.C., A. T. and E.M. contributed to the interpretation of the results. G.G., V.W., H.M., M.C. and F.C. wrote the manuscript. All authors contributed to the discussions and in improving the writing of the manuscript.

## Funding

## Competing interests

The authors declare no competing interests.
