## [Transparent Peer Review file · Communications Chemistry]

Real-Time Tracking of the intramolecular vibrational dynamics of liquid water

Corresponding Author: Professor Francesca Calegari

This manuscript has been previously reviewed at another journal. This document only contains information relating to versions considered at Communications Chemistry.

Version 0:

Reviewer comments:

Reviewer #1

(Remarks to the Author)

Most of my prior comments have been addressed in this revised manuscript; however, the following points still need revision:

Issue (7) still has problems (regarding the description of their vibrational frequency model.)

The authors extracted a Raman spectrum from Darwin's paper in *The Analyst* and fit it to 3 components, but the figure in that paper shows a fit to 5 different peaks and the associated text says "The application of Gaussian functions in the deconvolution procedure for different hydrogen bound OH, e.g. amount of functions and their positions, varies in the publications." It seems that the authors of that paper would argue that there are disputes about this assignment. Hence, the authors of this paper should emphasize that their fit is not simply the consensus of what is known about the spontaneous Raman spectrum, but actually a particular approximation that is chosen for their analysis. At a minimum, they need to say something like "There is a good deal of debate about the specific assignment of the OH Raman band, but we found that we can fit our results with a simple 3-peak fit to existing spectra."

Also, they should describe how sensitive their fitting is to this particular model. It seems like the 25-fs decoherence could be modeled by any extremely broad vibrational lineshape that was at approximately 3200 cm⁻¹. Do reasonable changes in their fit to the CW Raman spectrum affect the modeled fs results? If the choice of model really does not affect their results, that should be included in the text. For instance, what if they just fit the whole water band to a single Gaussian function; could they still model their data with this?

(11) (Regarding the comparison of their FT spectra to the CW Raman spectrum of water.) I think this still needs to be addressed in the text because many people in the field expect the FT of the coherent oscillations to generate a Raman spectrum of the sample. The authors should include a few sentences about why that is not appropriate given the current conditions.

Reviewer #2

(Remarks to the Author)

The authors addressed my point. I think that the manuscript is now good for publication.

Version 1:

Reviewer comments:

Reviewer #1

(Remarks to the Author)

The authors have adequately addressed all of my prior concerns. I now consider the manuscript acceptable for publication without any further changes.

Reviewer#1 (Remarks to the Author):

Most of my prior comments have been addressed in this revised manuscript; however, the following points still need revision:

Issue (7) still has problems (regarding the description of their vibrational frequency model.) The authors extracted a Raman spectrum from Darwin's paper in *The Analyst* and fit it to 3 components, but the figure in that paper shows a fit to 5 different peaks and the associated text says "The application of Gaussian functions in the deconvolution procedure for different hydrogen bound OH, e.g. amount of functions and their positions, varies in the publications." It seems that the authors of that paper would argue that there are disputes about this assignment. Hence, the authors of this paper should emphasize that their fit is not simply the consensus of what is known about the spontaneous Raman spectrum, but actually a particular approximation that is chosen for their analysis. At a minimum, they need to say something like "There is a good deal of debate about the specific assignment of the OH Raman band, but we found that we can fit our results with a simple 3-peak fit to existing spectra." Also, they should describe how sensitive their fitting is to this particular model. It seems like the 25-fs decoherence could be modeled by any extremely broad vibrational lineshape that was at approximately 3200 cm^{-1} . Do reasonable changes in their fit to the CW Raman spectrum affect the modeled fs results? If the choice of model really does not affect their results, that should be included in the text. For instance, what if they just fit the whole water band to a single Gaussian function; could they still model their data with this?

We thank the reviewer for this comment. Following their suggestion, we modified the text in the Supplementary Note 6 of the SI as follows: "Although there is no general consensus about the specific number of gaussian components to be used to fit the OH stretch Raman band, a minimal model based on three components can be used to properly describe both spontaneous Raman and time-domain ISRS spectra." This is confirmed in the following figure, showing that the spontaneous Raman spectrum reported in the literature [*The Analyst* 141, 6329–6337 (2016)] can be well approximated with a single gaussian and excellently reproduced with three components:

Fit of the Spontaneous Raman spectrum in [*The Analyst* 141, 6329–6337 (2016)] (dots) performed using a single gaussian (blue solid line) or three gaussian components (red solid line).

In our analysis, we used the deconvolution of the CW Raman spectrum as a starting point for the fit, keeping the minimum number of components required to correctly model the time domain data. Based on this argument, we satisfactorily reproduced the trace in Figure 3a of the main manuscript using a single damped cosine function with a 25-fs time constant and oscillation frequency of 3335 cm^{-1} . Instead, for the maps in Figure 4 of the main manuscript we included three components. This leads to a better description of the data, which are two-dimensional and thus have a richer information content than the time trace in Figure 3a. As shown in the figure below, under these conditions the agreement with ISRS traces at selected probe wavelengths slightly improves (the residual value decrease from 0.0450 to 0.0445 when going from a single gaussian to three gaussians), while including five gaussians would result in overfitting. We clarified this concept in the main text: “The frequencies, intensities, dephasing values of the modes are extracted by fitting the spontaneous Raman spectrum of liquid water [*The Analyst* 141, 6329–6337 (2016)] with the minimum number of components that correctly reproduce both spontaneous Raman and ISRS data (see Supplementary Note 6); the inhomogeneous spectral broadening contributions are modelled as gaussian decays.”

Slices of the 2D ISRS map extracted at different wavelengths for the three chirp conditions C1, C2 and C3, using frequency, intensity and dephasing values obtained by fitting the Spontaneous Raman spectrum of the OH stretch using a single gaussian (dashed lines) and three gaussian (solid lines) components.

(11) (Regarding the comparison of their FT spectra to the CW Raman spectrum of water.) I think this still needs to be addressed in the text because many people in the field expect the FT of the

coherent oscillations to generate a Raman spectrum of the sample. The authors should include a few sentences about why that is not appropriate given the current conditions.

"We thank the reviewer for the comment. We note that spontaneous Raman (SR) spectroscopy is an incoherent process, as light couples with thermally activated vibrational modes. In contrast, the ISRS probes vibrational coherences stimulated by the pump and read out by the broadband probe through multiple Raman transitions (see Fig. 2d-e of the manuscript), making the relative intensity of different Raman modes sensitive to the probe chirp. In addition, the pump duration can modulate the relative intensity of different Raman modes -suppressing those with vibrational periods longer than the pump. For these reasons, SR and ISRS spectra cannot be directly compared. We clarified this aspect in the Results section of the manuscript: "We remark that the spontaneous Raman and ISRS cross sections depend differently on the thermal population of a given mode, due to the coherent nature of the latter [Phys. Rev. Lett. 133, 206902 (2024)]. Moreover, the ISRS intensity ratios are modulated by the temporal and spectral properties of the beams, and may differ from the spontaneous Raman spectrum [J. Phys. Chem. A 2015, 119, 36, 9506–9517]."